# Diagnostic accuracy of Truenat Tuberculosis and Rifampicin-Resistance assays in Addis Ababa, Ethiopia

Abyot Meaza[1,2]*, Ephrem Tesfaye[1], Zemedu Mohamed[1], Betselot Zerihun[1], Getachew Seid[1], Kirubel Eshetu[1], Miskir Amare[1], Waganeh Sinshaw[1], Biniyam Dagne[1], Hilina Mollalign[1], Getu Diriba[1], Melak Getu[1], Bazezew Yenew[1], Mengistu Tadesse[1], Dinka Fikadu[1], Yeshiwork Abebaw[1], Shewki Moga[1], Abebaw Kebede[1], Habteyes Hailu Tola[1], Ayinalem Alemu[1], Muluwork Getahun[1], Balako Gumi[2]

1 Ethiopian Public Health Institute (EPHI), Addis Ababa, Ethiopia, 2 Aklilu Lemma Institute of Pathobiology, Addis Ababa University, Addis Ababa, Ethiopia

* abimeaza@gmail.com

**Data Availability Statement:** All relevant data are within the manuscript.

**Funding:** This study was funded by Foundation for Innovative New Diagnostics (FIND), which receives

## Abstract

### Background

Rapid and sensitive Tuberculosis (TB) diagnosis closer to patients is a key global TB control priority. Truenat assays (MTB, MTB Plus, and MTB-RIF Dx) are new TB molecular diagnostic tools for the detection of TB and Rifampicin (RIF)-resistance from sputum samples. The diagnostic accuracy of the assays is needed prior to implementation in clinical use in Ethiopia. This study aimed to determine the sensitivity and specificity of Truenat assays; and aimed to compare the assays to the Xpert MTB/RIF assay.

### Methods

A prospective evaluation study was conducted among 200 presumptive TB patients in microscopy centers in Addis Ababa, Ethiopia from May 2019 to December 2020. Culture (Solid and Liquid methods) and phenotypic (liquid method) drug susceptibility testing (DST) were used as a reference standard.

### Results

Of 200 adult participants, culture confirmed TB cases were 25 (12.5%), and only one isolate was resistant to RIF by phenotypic DST. The sensitivity of Truenat MTB was 88.0% [95% CI 70.1, 95.8], while 91.7 [95% CI 74.2, 97.7] for Truenat MTB Plus at the microscopy centers. The specificity of Truenat MTB was 97.2% [95% CI 93.1, 98.9], while for Truenat MTB Plus was 97.2% [95% CI 93.0, 99.0]. The sensitivity of Truenat MTB was 90.5% while for MTB Plus, 100% compared to the Xpert MTB/RIF assay.

### Conclusion

Truenat assays were found to have high diagnostic accuracy. The assays have the potential to be used as a point of care (POC) TB diagnostic tests.

funding from the Bill & Melinda Gates Foundation [grant no. OPP1208706]. The funder had no role in study design, data collection, data analysis, data interpretation, or writing of the manuscript. The corresponding author had full access to all the data in the study and had final responsibility for the decision to submit for publication.

**Competing interests:** The authors have declared that no competing interests exist.

## Introduction

Tuberculosis (TB) is causing morbidity and mortality among millions each year across the world [1]. The World Health Organization (WHO) recent report indicated 10 million new cases and 1.4 million deaths occurred globally in 2019 [1]. It is a major public health problem in Ethiopia with an estimated incidence of 140 per 100,000 in 2019 [1]. Ethiopia remains one of the countries with high TB, TB/HIV and Multi-drug resistant-TB (MDR-TB) burden [1]. Although TB incidence has been declining steadily over the past years, available information indicates that only two-thirds of estimated cases are being detected [2]. This indicates, a considerable proportion of TB cases are not diagnosed. As a result, rapid and sensitive TB diagnostic tools that increase TB and RIF resistance detection are required to achieve end TB targets.

The use of rapid molecular tests is increasing across the world, and many countries are phasing out the use of smear microscopy for diagnostic purposes [3]. However, microscopy remains a useful diagnostic tool for treatment monitoring. Moreover, despite advances in diagnostics, a considerable proportion of TB cases reported to WHO are still clinically diagnosed instead of bacteriologically confirmed [3].

Rapid diagnosis and timely initiation of treatment are essential to stop the transmission of TB disease. However, an estimated 4.1 million TB cases are left undiagnosed globally each year, leading to prolonged transmission of the disease in the population [1]. In most countries, smear microscopy remains the only option for the diagnosis of TB, though it detects only 45% of infections [4]. For these reasons, new rapid molecular TB diagnostic tools that can be instituted at the microscopy diagnostic centers as point of care testing are a research and implementation priority [4].

A new molecular TB diagnostic tool named Truenat which includes Truenat MTB, Truenat MTB Plus, and Truenat MTB-RIF Dx assays has been developed recently by Molbio Diagnostics, Bangalore, India. The Truenat assays utilize chip-based real-time micro-polymerase chain reaction (PCR) for detection of TB and RIF-resistance from Deoxyribonucleic Acid (DNA) that is extracted from sputum sample within an hour [5–7]. The sensitivity and specificity of the new assays have not been studied and are needed prior to implementation in clinical use in Ethiopia. Therefore, this study was aimed to determine the sensitivity and specificity of Truenat MTB assay and MTB Plus assay, in the detection of TB and Truenat MTB-RIF Dx assay for RIF resistance detection in the microscopy centers in Addis Ababa, Ethiopia.

## Materials and methods

A prospective cross-sectional study was conducted in three selected microscopy centers in Addis Ababa, Ethiopia from May 15, 2019, to December 08, 2019. The sites are selected based on their number of sputum smear examination, number of smear positivity rate and unavailability of GeneXpert. The study enrolled 200 [8] adults with clinical suspicion of pulmonary TB who were 18 years or older, willing to provide four rounds of sputum samples and willing to provide informed consent as inclusion criteria. The sensitivity and specificity of Truenat assays for TB detection and RIF-resistance detection from sputum samples was determined using phenotypic culture (LJ and MGIT) and DST (MGIT) methods as a reference standard. The Xpert MTB/RIF assay was used as a comparator standard.

### Enrolment procedures

Four rounds of sputum samples were collected from each eligible study participant, two spot sputum samples in the first-day visit and two sputum samples (one morning, and one spot) in the second-day visit. Direct smear microscopy and Truenat MTB, MTB Plus assays, and MTB-RIF Dx assay were performed from direct sputum samples at the microscopy centers.

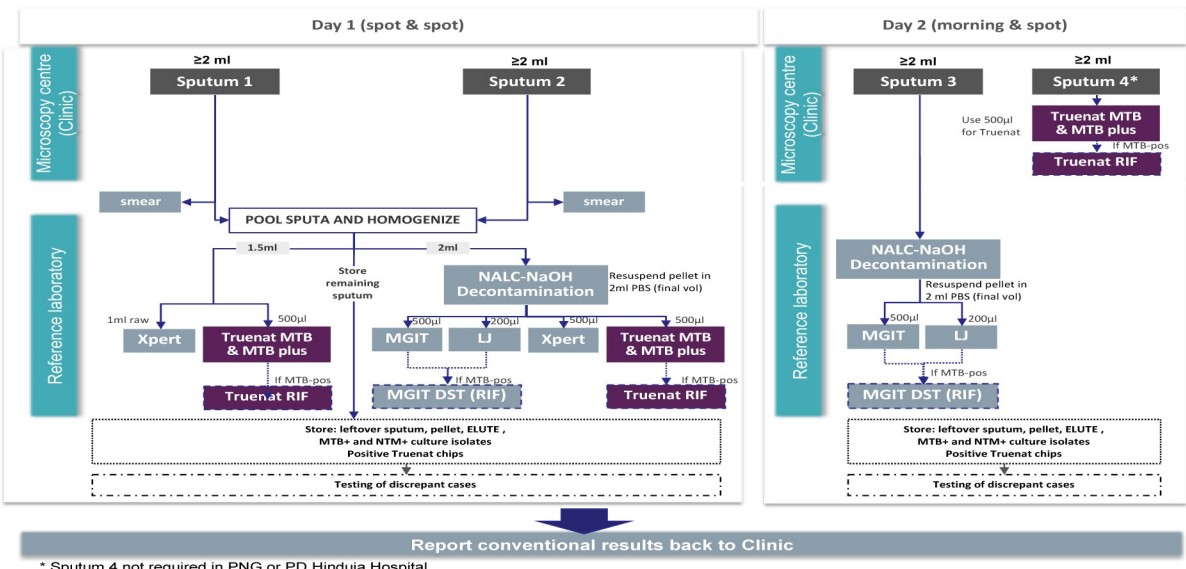

**Fig 1. Laboratory testing workflow at microscopy center and reference laboratory.**

Truenat assays, GeneXpert testing, culture, and phenotypic DST were performed at the reference laboratory (Fig 1) [5, 9–11].

## Truenat testing steps

Two drops of liquefaction buffer added to 0.5ml sputum and incubated for five minutes at room temperature to allow the sample to liquefy. Then, 0.5 mL of the liquefied sputum transferred to a lysis buffer bottle using the transfer pipette and two more drops of the liquefaction buffer added to the lysis buffer bottle. The cap closed tightly and mixed well. Then, the lysis buffer bottle incubated at room temperature for three minutes. Finally, the entire content of the lysis buffer bottle transferred to the cartridge's sample chamber using the transfer pipette and directly loaded onto the Truepep Auto chip interface to extract MTB DNA. The extracted DNA was transferred to the Truenat MTB or Truenat MTBPlus chip, and then onto the Truelab PCR machine, which detects the presence of MTB DNA. Then the Truelab machine provides the automatic results as either MTB-detected, MTB not detected or indeterminate. For MTB positive results, another aliquot of the same DNA extract was transferred (reflex) to the Truenat MTB-RIF Dx chip [5–7].

## Ethical approval and consent

The study was approved by the scientific and ethical review office of EPHI (approval number: EPHI-IRB-139-2017). Informed written consent was obtained from each study participants.

## Statistical analysis

Data was coded and double entered in to the OpenClinica online statistical tool (version 3.4, USA) [12]. Data was exported to SPSS version 23 for data checking, cleaning and analysis. The sensitivity and specificity of Truenat assays were estimated with 95% confidence intervals (CIs). Results were reported as being statistically significant whenever the p-value is less than 0.05.

## Results

### Sociodemographic and clinical characteristics

In the current study, 200 participants were included in the final analysis. Participants ranged in age from 18 to 81 years old, with a mean age of 39.8 (±14.7) years. Among the participants, 100 (50%) were male and twenty (10%) participants had previous TB history. Of the total participants, 25 (12.5%) participants were bacteriologically-confirmed TB cases and only one (1/25) was MTB-RR case based on phenotypic DST. Among bacteriologically-confirmed cases, 18 were smear-positive while seven were smear-negative.

### Diagnostic accuracy of the Truenat MTB, MTBPlus and MTB-RIF Dx assays

The sensitivity of Truenat MTB was 88.0% [95% CI 70.1, 95.8], while 91.7% [95% CI 74.2,97.7] for MTB Plus (Table 1). The specificity of Truenat MTB and MTB Plus were similar, 97.2% [95% CI 93.1,98.9]. The sensitivity of Truenat MTB and Truenat MTB Plus assays for smear-negative and culture-positive specimens was 42.9% N = 7 [95% C 15.8, 74.9] and 50% N = 6 [95% CI 18.8, 81.2] respectively (Table 1). The diagnostic accuracy of Truenat assays was also evaluated from direct pooled sputum specimens in the reference laboratory. The diagnostic accuracy showed lower sensitivity for Truenat MTB and higher sensitivity for MTB Plus assay in the reference laboratory than microscopy centers (Sensitivity difference for MTB = -1.6% [95% CI: -3.4, -0.5], Sensitivity difference for MTB plus = +8.3% [95% CI: +9, 2.3]) (Table 1).

### Comparison of the diagnostic accuracy of Truenat assay to Xpert MTB/RIF

Among samples with valid Truenat, Xpert and culture results, the sensitivity of Truenat MTB was 90.5% [95% CI 71.1, 97.4], while 100% [95% CI 83.2, 100] for Truenat MTB Plus compared to Xpert MTB/RIF. The specificity of Truenat MTB and MTB Plus was approximately similar compared to Xpert. Among samples with valid Truenat MTB-RIF Dx assay, Xpert MTB/RIF, and phenotypic DST, the sensitivity and specificity of the MTB-RF Dx assay were similar (100%) compared to Xpert MTB/RIF (Fig 2).

### Non-determinate results for Truenat and Xpert

Non-determinate or invalid results of Truenat assay and Xpert MTB/RIF from direct sputum specimens received in the reference laboratory were shown in Table 2. The proportion of Trueprep non-determinate results or unsuccessful extraction in direct sputum specimen was

**Table 1. Diagnostic accuracy of the Truenat assays for MTB detection and MTB-RIF resistance detection among 200 enrolled participants.**

| Truenat assays | N | TP | FP | FN | TN | Sen% [95%CI] | Spe % [95%CI] | Sen %, SN&CP T[95%CI] | Spe %, SP&CP [95%CI] |
|---|---|---|---|---|---|---|---|---|---|
| Microscopy center–Direct | | | | | | | | | |
| Truenat MTB | 169 | 22 | 4 | 3 | 140 | 88.0 [70.1, 95.8] | 97.2 [93.1,98.9] | 3/7 (42.9) [15.8, 74.9] | 100 N = 18 [82.4,100] |
| Truenat MTBPlus | 166 | 22 | 4 | 2 | 138 | 91.7 [74.2,97.7] | 97.2 [93.0,99.0] | 3/6 (50) [18.8, 81.2] | 100 N = 18 [82.4,100] |
| Truenat MTB-RIF | 19 | 1 | 0 | 0 | 18 | 100 [5.5,100] | 100 [78.1,100] | 100 N = 7 | 100 N = 18 [81.6,100] |
| Reference lab–Direct | | | | | | | | | |
| Truenat MTB | 153 | 19 | 1 | 3 | 130 | 86.4 [66.7, 95.3] | 99.2 [95.8,99.9] | 2/6(33.3) [9.7, 70.0] | 100 N = 18 [81.6,100] |
| Truenat MTBPlus | 153 | 19 | 2 | 0 | 134 | 100 [83.2,100] | 98.5 [81.6,100] | 3/7(42.9) [15.8, 75.0] | 100 N = 18 [81.6,100] |
| Truenat MTB-RIF | 18 | 1 | 0 | 0 | 17 | 100 [5.5,100] | 100 [81.6,100] | 0 N = 7 | 100 N = 18 [81.6,100] |
| Reference lab–Decontaminated | | | | | | | | | |
| Truenat MTB | 161 | 18 | 1 | 5 | 137 | 78.3 [66.7, 95.3] | 99.3 [97.1,100] | 1/7 (14.3) [2.6, 51.3] | 100 N = 18 [81.6,100] |
| Truenat MTBPlus | 148 | 19 | 0 | 3 | 126 | 86.4 [66.7, 95.3] | 99.3 [96.2,99.9] | 2/6 (33.3) [9.7, 70.0] | 100 N = 18 [81.6,100] |
| Truenat MTB-RIF | 18 | 1 | 0 | 0 | 17 | 100 [5.5,100] | 100 [81.6,100] | 0 N = 7 | 100 N = 18 [80.6,100] |

a,

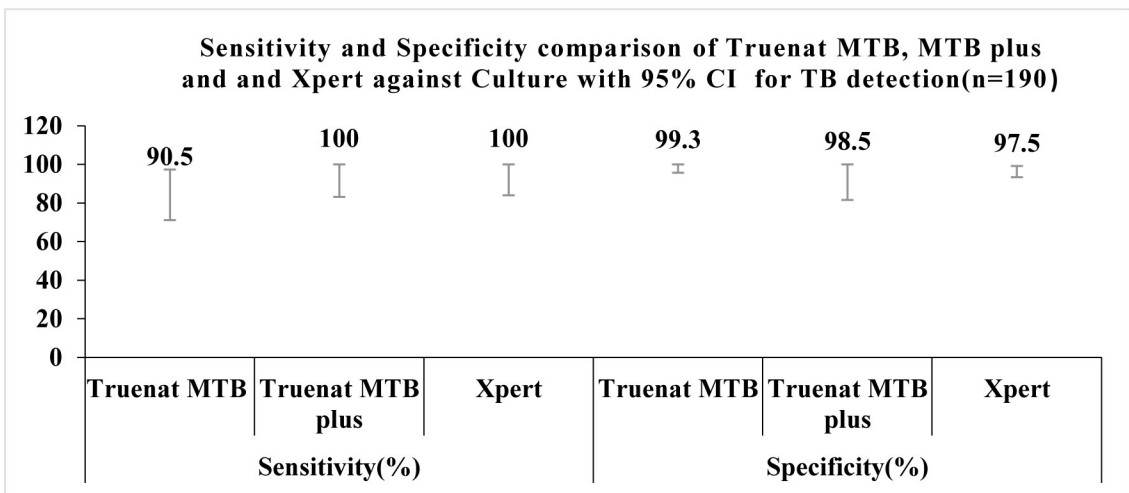

b,

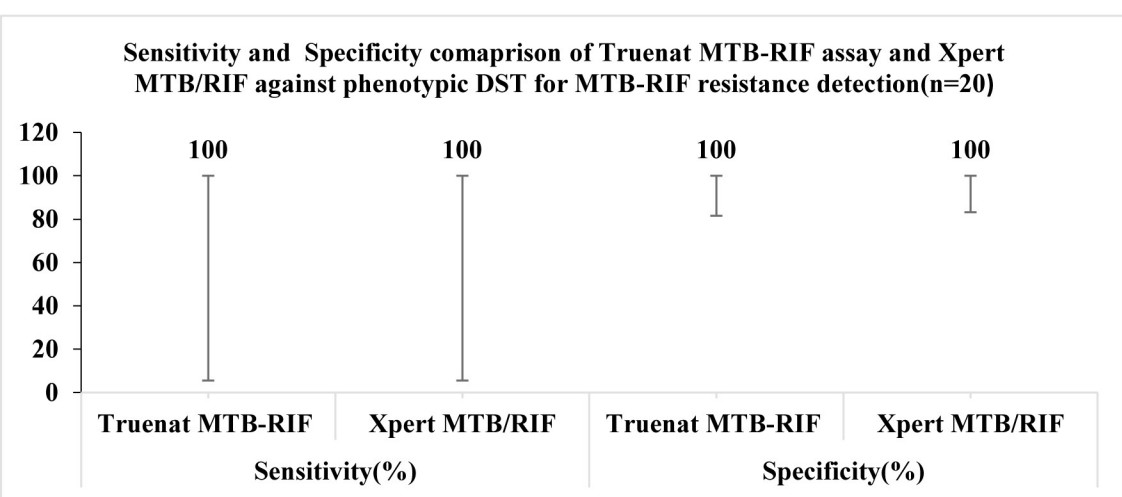

**Fig 2.** Diagnostic accuracy comparison of Truenat assays and Xpert MTB/RIF against Cultuer/DST for MTB detection (a) and MTB-RIF resistance detection (b).

**Table 2. Non-determinate results of Truenat assay and Xpert MTB/RIF in direct sputum specimen received in the reference laboratory.**

| Test | All results | Valid results | Invalid results | Invalid rate, % |
|---|---|---|---|---|
| Truenat MTB | 190 | 169 | 21 | 11.1 |
| Truenat MTBPlus | 190 | 155 | 35 | 18.4 |
| Truenat MTB-RIF | 20 | 18 | 2 | 10 |
| Truenat assays, Total | 400 | 342 | 58 | 14.5 |
| Xpert MTB/RIF | 190 | 183 | 7 | 3.7 |

6% (12/200). The non-determinant results of Trueprep extraction were valid after the procedure was repeated. The non-determinate rate of Truenat MTB was 11.1% (21/190), while 18.4% (35/190) for MTB plus assay. However, non-determinate results were resolved after repeating the test. The non-determinate rate for the Truenat MTB-RIF Dx assay was 10% (2/20) (Table 2). The overall non-determinate rates of Truenat assays were 14.5% (58/400), whereas 3.7% (7/190) for Xpert MTB/RIF assay from direct sputum specimens in a reference laboratory (Table 2).

## Discussion

In the current study, we evaluated the diagnostic accuracy of Truenat assays (Truenat MTB, Truenat MTB Plus and Truenat MTB-RIF Dx) for the diagnosis of TB and RIF-resistance in smear microscopy centers found in Addis Ababa, Ethiopia. The sensitivity of Truenat MTB was 88.0% [95% CI 70.1, 95.8], while 91.7% [95% CI 74.2,97.7] for MTB Plus. The specificity of Truenat MTB and MTB Plus were similar, 97.2% [95% CI 93.1,98.9]. The sensitivity of Truenat MTB was 90.5% [95% CI 71.1, 97.4], while 100% [95% CI 83.2, 100] for Truenat MTB Plus compared to Xpert MTB/RIF. Similar sensitivity and specificity (100%) of Truenat MTB-RIF Dx assay were found compared to Xpert MTB/RIF assay.

The sensitivity of the Truenat MTB was comparable and higher for MTB Plus in the reference laboratory compared to the microscopy centers. The specimens used at the microscopy centers were single spot specimens, whereas pooled sputum specimens were used in the reference laboratory, which might increase the sensitivity due to the high specimen bacillary load in the pooled specimens. The assays showed high specificity and there was no significant difference between specificity in the microscopy centers and reference laboratory. Similar findings were also observed in other studies conducted in other study settings [13, 14].

The sensitivity of Truenat assays for TB detection in smear-negative and culture-positive specimens was low for both Truenat MTB (42.9%) and Truenat MTB Plus (50%) assays. The present study sensitivity is lower than the previous study finding (86.2%). This is most probably due to sample size difference. In the previous study, the valid sample size was relatively larger (58) [14] than the current study (only seven).

The sensitivity and specificity of Truenat MTB-RIF Dx assay was high for the test done from direct sputum samples both in the microscopy centers and a reference laboratory. The current study findings are slightly higher than the results presented in the recent WHO rapid communication document on Truenat diagnostic accuracy [15]. However, our estimate of Truenat MTB-RIF Dx performance to detect RIF-resistance is uncertain due to the small valid sample size for the test at both microscopy centers and the reference laboratory. In microscopy centers, there were 19 valid results of which 18 were sensitive to RIF and the remaining one was RIF-resistant. Of 18 valid results in the reference laboratory, 17 were sensitive to RIF while the remaining one was RIF-resistant.

The diagnostic accuracy of Truenat assay was comparable with Xpert MTB/RIF for TB detection from direct sputum samples in the reference laboratory. The sensitivity was lower (90.5%) for Truenat MTB, and it was similar (100%) for Truenat MTB Plus compared to the Xpert MTB/RIF assay. This might be due to the high quality and increased limit of detection (LOD) of Truenat MTB Plus micro-PCR chips which increased its performance, (LOD for Truenat MTB, 100cfu/ml; for Truenat MTB Plus, 30cfu/ml) [6]. High diagnostic accuracy has been reported from India [14], which is comparable with the current study finding. The specificity of Truenat MTB and MTB Plus was similar when compared with Xpert MTB/RIF assay. Among participant specimens in the MTB-RIF resistance detection with valid results, sensitivity and specificity of Truenat MTB-RIF Dx were similar, 100% with Xpert MTB/RIF assay.

The diagnostic accuracy of Truenat assays in decontaminated (sediment) sputum specimens was lower compared to direct sputum specimens. Decontaminated specimens that are more concentrated and likely to increase the performance of the assay showed lower performance in the study. This might need further study with an increased sample size. However, improved performance was obtained from the Xpert MTB/RIF assay on sputum sediment samples in a study done in Tanzania [16]. Decontamination procedures which are costly and time-consuming couldn't increase the diagnostic accuracy in the current study. This finding indicated that Truenat assays with direct sputum specimens have the potential to be used in primary health settings.

The overall non-determinate rate of Truenat assays was high (14.5%) compared to the invalid rate of Xpert MTB/RIF assay (3.7%) from direct sputum samples. These findings are in line with the early Xpert evaluation studies [17, 18] in which high Xpert invalid rates were reflected. However, since the current comparison of Truenat assays and Xpert MTB/RIF was conducted in the same setting, the proportion of non-determinate results for Truenat assays was high. Upon the first repeated testing, the majority of the non-determinate results were resolved. However, repeating the test of the assay increases the cost and turnaround time.

## Conclusion

The diagnostic accuracy of Truenat assays was found to be high for MTB detection and MTB-RIF resistance detection in microscopy centers. The assays have the potential to be used as POC TB diagnostic tests. The diagnostic accuracy of Truenat assays was comparable with the Xpert MTB/RIF assay in the direct sputum samples in the reference laboratory. The current diagnostic accuracy estimate of the Truenat MTB-RIF Dx assay in our study was uncertain due to the small sample size. Therefore, we recommend further study with a larger sample size. The non-determinate rate for Truenat assays was high; hence, we recommend that the manufacturer should improve the assays to reduce the non-determinate results.

## Acknowledgments

The authors would like to thank the study participants for volunteer participation in this study. We extend our gratitude to study sites for their collaboration and contribution to this study.

## Author Contributions

**Conceptualization:** Abyot Meaza.

**Data curation:** Abyot Meaza, Zemedu Mohamed, Dinka Fikadu, Habteyes Hailu Tola.

**Formal analysis:** Abyot Meaza, Zemedu Mohamed, Dinka Fikadu, Habteyes Hailu Tola.

**Funding acquisition:** Abyot Meaza, Ephrem Tesfaye, Muluwork Getahun.

**Investigation:** Betselot Zerihun, Getachew Seid, Miskir Amare, Waganeh Sinshaw, Biniyam Dagne, Hilina Mollalign, Getu Diriba, Melak Getu, Bazezew Yenew, Yeshiwork Abebaw, Shewki Moga.

**Methodology:** Abyot Meaza, Zemedu Mohamed, Kirubel Eshetu, Dinka Fikadu, Abebaw Kebede, Habteyes Hailu Tola, Balako Gumi.

**Project administration:** Ephrem Tesfaye, Getachew Seid, Miskir Amare, Ayinalem Alemu, Muluwork Getahun.

**Resources:** Abyot Meaza, Ephrem Tesfaye, Miskir Amare, Getu Diriba, Mengistu Tadesse, Ayinalem Alemu.

**Software:** Zemedu Mohamed, Dinka Fikadu.

**Supervision:** Abyot Meaza, Ephrem Tesfaye, Zemedu Mohamed, Betselot Zerihun, Kirubel Eshetu, Mengistu Tadesse, Habteyes Hailu Tola, Balako Gumi.

**Validation:** Ephrem Tesfaye, Zemedu Mohamed, Betselot Zerihun, Kirubel Eshetu, Mengistu Tadesse, Dinka Fikadu, Abebaw Kebede, Habteyes Hailu Tola, Balako Gumi.

**Visualization:** Zemedu Mohamed, Betselot Zerihun, Kirubel Eshetu, Balako Gumi.

**Writing – original draft:** Abyot Meaza.

**Writing – review & editing:** Abyot Meaza, Ephrem Tesfaye, Zemedu Mohamed, Kirubel Eshetu, Getu Diriba, Yeshiwork Abebaw, Abebaw Kebede, Habteyes Hailu Tola, Ayinalem Alemu, Balako Gumi.

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
