## [Decision Letter · Decision Letter 0]

29 Oct 2021

PONE-D-21-22469Diagnostic accuracy of Truenat Tuberculosis and Rifampicin-Resistance assays in Addis Ababa, EthiopiaPLOS ONE

Dear Meaza,

Thank you for submitting your manuscript to PLOS ONE. After careful consideration, we feel that it has merit but does not fully meet PLOS ONE’s publication criteria as it currently stands. Therefore, we invite you to submit a revised version of the manuscript that addresses the points raised during the review process.

As highlighted by two expert reviewers, this study is well executed and well written. Once the authors  addrees  comments from the reviewers, this manuscript can be accepted for publication.

We look forward to receiving your revised manuscript.

Kind regards,

Pradeep Kumar, Ph.D.

Academic Editor

PLOS ONE

2. Thank you for stating the following in the Acknowledgments/Funding Section of your manuscript:

“This study was funded by Bill & Melinda Gates Foundation (OPP1208706). The funder had no role in study design, data collection, data analysis, data interpretation, or writing of the manuscript. The corresponding author had full access to all the data in the study and had final responsibility for the decision to submit for publication.”

“This study was funded by Bill & Melinda Gates Foundation (OPP1208706). The funder had no role in study design, data collection, data analysis, data interpretation, or writing of the manuscript. The corresponding author had full access to all the data in the study and had final responsibility for the decision to submit for publication.”

“We declare that there is no any conflict of interest that might have influenced the performance and the presentation of the work described in this manuscript.”

4. Please include a caption for figure 2.

5. Please upload a copy of Figure 1-3, to which you refer in your manuscript. If the figure is no longer to be included as part of the submission please remove all reference to it within the text.

Reviewers' comments:

Reviewer's Responses to Questions

**Comments to the Author**

1. Is the manuscript technically sound, and do the data support the conclusions?

Reviewer #1: Yes

Reviewer #2: Yes

2. Has the statistical analysis been performed appropriately and rigorously? 

Reviewer #1: Yes

Reviewer #2: Yes

3. Have the authors made all data underlying the findings in their manuscript fully available?

Reviewer #1: Yes

Reviewer #2: No

4. Is the manuscript presented in an intelligible fashion and written in standard English?

Reviewer #1: Yes

Reviewer #2: Yes

5. Review Comments to the Author

Reviewer #1: It was a pleasure to review the manuscript by Meaza et al. on the accuracy of the Truenat TB & RIF resistance assays in Addis Ababa, Ethiopia. Their study results contribute to the important discourse on the increasing need to promptly detect tuberculosis and drug resistance to allow the institution of appropriate management measures early. The manuscript is well written, the methods and reference standards are adequate, as are the results and discussion. The authors also briefly alluded to the limitations of the study. Below are some comments that the authors may wish to address:

1. Line 57: change ‘Truent’ to “Truenat”

2. Line 58: add “s” to sample to read “samples”

3. Line 104: change ‘...assays is not studied...’ to “assays have not been studied” as appropriate.

4. Line 110: How many and how were microscopy centers selected? Also, how were the participants selected (inclusion criteria) and recruited, e.g., consecutively, or randomly?

5. Lines 185-188: Why were pooled samples only analyzed at the reference laboratory and not at the microscopy centers as well? Doing so would have provided extra information on the diagnostic proficiency at the microscopy centers.

6. Lines 192-193: Can the authors consider deleting “slightly” considering it may be a stretch to characterize the difference between 50% and 86.2% as slight?

7. Lines 238-240: This conclusion may be presumptuous because data reported herein does not demonstrate the positivity levels of the ZN slides, e.g., 1+, 2+ or 3+? Including this data would have bolstered the author’s assertion. Can the authors comment on this or adjust their conclusion as appropriate?

8. If the intention of the authors is to introduce the use of Truenat tests widely in Ethiopia, did the authors consider using sentinel sites throughout the Country instead of the capital city alone?

Reviewer #2: The manuscript by Abyot Meaza et al., entitled “Diagnostic accuracy of Truenat Tuberculosis and Rifampicin- Resistance assays in Addis Ababa, Ethiopia” report the comparative analysis on sensitivity and specificity between the recently introduced Trenat-Tb assays with the previously known XpertTB assay. As per the data collected from 200 patients from Ethiopia, two of the three Truenat assay platforms (MTB Plus and MTB-RIF Dx) were found to be highly comparable to XpertTb assay in terms of specificity and sensitivity. The authors suggest the potential of Truenat assays in future point of care TB diagnostic test while accepting the limitation associated with the current small scale study. The study also recommends further standardization in reducing the relatively high rate of non-determinate or invalid results in Truenat assays.

Although a short report, the manuscript is well written and provided sufficient statistical analysis when required. Still, there are some modifications needed for the better representation of the data.

Major comments

1) The authors may consider providing a graphical description or a simple flow chart of the method as a separate figure, before describing the data.

2) The methods are not well explained. Please add details on the sputum processing, volume and proportion with the liquefaction buffer etc.

3) The data tables are clear however, the figure may be replotted for a better representation.

Minor comments

1) Line 57, 184: Correct the spelling of “Truent”

2) Line 91: remove “highly”

3) Line 102-104: Combine the sentences.

6. PLOS authors have the option to publish the peer review history of their article (what does this mean?). If published, this will include your full peer review and any attached files.

Reviewer #1: **Yes: **Isdore Chola Shamputa

Reviewer #2: **Yes: **Jees Sebastian

---

## [Author Response · Author response to Decision Letter 0]

9 Nov 2021

Reviewer #1: It was a pleasure to review the manuscript by Meaza et al. on the accuracy of the Truenat TB & RIF resistance assays in Addis Ababa, Ethiopia. Their study results contribute to the important discourse on the increasing need to promptly detect tuberculosis and drug resistance to allow the institution of appropriate management measures early. The manuscript is well written, the methods and reference standards are adequate, as are the results and discussion. The authors also briefly alluded to the limitations of the study. Below are some comments that the authors may wish to address:

1. Line 57: change ‘Truent’ to “Truenat”

Response: Thank you. Corrected as indicated in line 60 in track changes.

2. Line 58: add “s” to sample to read “samples”

Response: Corrected as indicated in line 62 in track changes.

3. Line 104: change ‘...assays is not studied...’ to “assays have not been studied” as appropriate.

Response: Corrected as indicated in line 107 in track changes.

4. Line 110: How many and how were microscopy centers selected? Also, how were the participants selected (inclusion criteria) and recruited, e.g., consecutively, or randomly?

Response: We thank the reviewer for this comment. The sites are selected based on their number of sputum smear examination, number of smear positivity rate and unavailability of GeneXpert. 

The study enrolled 200 adults with clinical suspicion of pulmonary TB who were 18 years or older, willing to provide four rounds of sputum samples and willing to provide informed consent as inclusion criteria. Eligible study participants who were willing to participate in the study recruited consecutively.

Revised as indicated in line 114-119 in track changes. 

5. Lines 185-188: Why were pooled samples only analyzed at the reference laboratory and not at the microscopy centers as well? Doing so would have provided extra information on the diagnostic proficiency at the microscopy centers.

Response: We thank the reviewer for this comment. 

Four sputum samples were collected from each eligible study participant, two spot sputum samples in the first-day visit and pooled for Truenat, Xpert, and Culture in the reference laboratory. Two sputum samples (one morning-for culture in the reference lab and one spot for Truenat in microscopy center) in the second-day visit.

Pooling of two sputum samples could not be performed in the microscopy center where the biosafety level of the laboratory is minimal (BSL 1). This procedure was performed in the reference laboratory in Biological safety cabinet (BSL 2 or 3)

The purpose of analyzing pooled samples only in the reference lab was to see the diagnostic accuracy of Truenat from the pooled samples over direct samples. The diagnostic accuracy showed lower sensitivity for Truenat MTB and higher sensitivity for MTB Plus assay in the reference laboratory than microscopy centers as indicated in line 166-170 in track changes.

6. Lines 192-193: Can the authors consider deleting “slightly” considering it may be a stretch to characterize the difference between 50% and 86.2% as slight?

Response: Yes, well appreciated for the suggestion and deleted as indicated in line 208 in track change.

7. Lines 238-240: This conclusion may be presumptuous because data reported herein does not demonstrate the positivity levels of the ZN slides, e.g., 1+, 2+ or 3+? Including this data would have bolstered the author’s assertion. Can the authors comment on this or adjust their conclusion as appropriate?

Response: We thank the reviewer for this comment.

 The study aimed to determine the sensitivity and specificity of Truenat assays in the detection of TB and RIF resistance detection in the microscopy centers. The reference standard and comparator standard are culture/DST and Xpert MTB/RIF respectively. Truenat assays were not planned to be compared to ZN smear microscopy as the sensitivity of sputum smear microscopy is less (60%, Umair M. Diagnostic Accuracy of Sputum Microscopy in Comparison With GeneXpert in Pulmonary Tuberculosis. Cureus. 2020 Nov 8;12(11):e11383. doi: 10.7759/cureus.11383). Microscopy centers were selected as the study area because they are the primary health settings where the Truenat assay planned to be evaluated and further implemented.

One of the challenges of Trueat assay in our study was high number of invalid rates. The overall non-determinate rate of Truenat assays was high (14.5%) compared to the invalid rate of Xpert MTB/RIF assay (3.7%) from direct sputum samples as indicated in table 2. This resulted in repeated testing which increases the cost and turnaround time.

Therefore we recommended that the manufacturer should improve the assays to reduce the non-determinate results.

8. If the intention of the authors is to introduce the use of Truenat tests widely in Ethiopia, did the authors consider using sentinel sites throughout the Country instead of the capital city alone?

Response: Yes we considered to implement and we appreciate for the comment. We considered 15 sentinel sites throughout the country to implement the testing and to see the operational feasibility of Truenat testing. For the purpose of evaluation study we used sites from the capital city alone which are near to the reference laboratory (for the reference and comparator standards) to transport the sample and to run the testing with the same day of sample collection.

Reviewer #2: The manuscript by Abyot Meaza et al., entitled “Diagnostic accuracy of Truenat Tuberculosis and Rifampicin- Resistance assays in Addis Ababa, Ethiopia” report the comparative analysis on sensitivity and specificity between the recently introduced Trenat-Tb assays with the previously known XpertTB assay. As per the data collected from 200 patients from Ethiopia, two of the three Truenat assay platforms (MTB Plus and MTB-RIF Dx) were found to be highly comparable to XpertTb assay in terms of specificity and sensitivity. The authors suggest the potential of Truenat assays in future point of care TB diagnostic test while accepting the limitation associated with the current small scale study. The study also recommends further standardization in reducing the relatively high rate of non-determinate or invalid results in Truenat assays.

Although a short report, the manuscript is well written and provided sufficient statistical analysis when required. Still, there are some modifications needed for the better representation of the data.

Major comments

1) The authors may consider providing a graphical description or a simple flow chart of the method as a separate figure, before describing the data.

Response: We thank for the reviewer comment. 

We provided laboratory testing workflow (sample flow, method of testing in reference laboratory or microscopy center) as figure 1.

2) The methods are not well explained. Please add details on the sputum processing, volume and proportion with the liquefaction buffer etc.

Response: Thank you for the comment.

It is well explained in detail as per the comment as indicated in line 131-137.

3) The data tables are clear however, the figure may be replotted for a better representation.

Response: Thank you for the comment.

The figure replotted as indicated in Figure 2.

Minor comments

1) Line 57, 184: Correct the spelling of “Truent”

 Response: corrected as indicated in line 60 and 199 in track changes.

2) Line 91: remove “highly”

Response: Removed as indicated in line 95 in track changes.

3) Line 102-104: Combine the sentences.

Response: combined as indicated in line 106-107in track changes.

---

## [Editor Report · Decision Letter 1]

24 Nov 2021

Diagnostic accuracy of Truenat Tuberculosis and Rifampicin-Resistance assays in Addis Ababa, Ethiopia

PONE-D-21-22469R1

Dear Meaza,

We’re pleased to inform you that your manuscript has been judged scientifically suitable for publication and will be formally accepted for publication once it meets all outstanding technical requirements.

Kind regards,

Pradeep Kumar, Ph.D.

Academic Editor

PLOS ONE
---

## [Editor Report · Acceptance letter]

14 Dec 2021

PONE-D-21-22469R1 

Diagnostic accuracy of Truenat Tuberculosis and Rifampicin-Resistance assays in Addis Ababa, Ethiopia 

Dear Dr. Meaza:

I'm pleased to inform you that your manuscript has been deemed suitable for publication in PLOS ONE. Congratulations! Your manuscript is now with our production department. 

Kind regards, 

on behalf of

Dr. Pradeep Kumar 

Academic Editor

PLOS ONE